

# The applications of anterior segment optical coherence tomography in glaucoma: a 20-year bibliometric analysis

Yijia Huang[1,2,*], Di Gong[1,2,*], Kuanrong Dang[1,2], Lei Zhu[1,2], Junhong Guo[1,2], Weihua Yang[1] and Jiantao Wang[1]

[1] Shenzhen Eye Hospital, Jinan University, Shenzhen Eye Institute, Shenzhen, China
[2] The First Affiliated Hospital of Jinan University, Jinan University, Guangzhou, China
* These authors contributed equally to this work.

Corresponding authors
Weihua Yang,
benben0606@139.com
Jiantao Wang,
jiantaowang65@163.com

## ABSTRACT

**Objective:** In the past 20 years, the research application of anterior segment optical coherence tomography (AS-OCT) in the field of glaucoma has become a hot topic, but there is still a lack of bibliometric reports on this scientific field. The aim of this study is to explore the research hotspots and trends in the field using bibliometric methods.

**Method:** Analyzing literature from 2004 to 2023 on AS-OCT in glaucoma within the SCI database, this study utilized Bibliometric, VOS viewer, and Cite Space for a comprehensive bibliometric analysis covering document counts, countries, institutions, journals, authors, references, and keywords.

**Results:** A total of 931 eligible articles were collected, showing a continuous increase in annual research output over the past 20 years. The United States, China, and Singapore were the top three countries in terms of publication volume, with 288, 231, and 124 articles, respectively, and there was close cooperation among these countries. The NATIONAL UNIVERSITY OF SINGAPORE, SUN YAT SEN UNIVERSITY, and SINGAPORE NATIONAL EYE CENTRE were the most productive institutions with 93, 92, and 87 articles, respectively. JOURNAL OF GLAUCOMA, INVESTIGATIVE OPHTHALMOLOGY & VISUAL SCIENCE, and OPHTHALMOLOGY were the journals with the highest number of publications, with 86, 69, and 46 articles, respectively. PROGRESS IN RETINAL AND EYE RESEARCH, published in the United States, was the top-cited journal. Researchers Aung Tin, He Mingguang, and David S. Friedman were highlighted for their contributions. The reference clustering was divided into 12 categories, among which "deep learning, anterior segment" were the most cited categories. The keywords of research frontiers include deep learning, classification, progression, and management.

**Conclusion:** This article analyses the academic publications on AS-OCT in the diagnosis and treatment of glaucoma over the last 20 years. Among them, the United States contributed the largest number of publications in this field, with the highest number of literature citations and mediated centrality. Among the prolific authors, aung, tin topped the list with 77 publications and 3,428 citations. Since the beginning of 2018, advances in artificial intelligence have shifted the focus of research in this field from manual measurements to automated detection and identification of relevant indicators.

# INTRODUCTION

Glaucoma is a leading cause of irreversible blindness worldwide. It is classified mainly into two types based on the anatomy of the anterior chamber angle: primary open-angle glaucoma (POAG) and primary angle-closure glaucoma (PACG) (*Jayaram et al., 2023*). By 2020, the estimated number of patients with POAG and PACG was projected to reach 59 million and 53 million, respectively. Notably, PACG carries a significantly higher risk of blindness than POAG, accounting for nearly half of all glaucoma-related blindness cases (*Quigley & Broman, 2006*; *Tham et al., 2014*; *Tso, Naumann & Zhang, 1998*). Recent advancements in ophthalmic technologies—such as intraocular pressure measurement, fundus photography, visual field testing, and optical coherence tomography (OCT)—have greatly enhanced the early diagnosis and management of glaucoma. A thorough understanding of the anterior segment's spatial structure is crucial in clinical practice. Thus, early qualitative and quantitative assessments of the anterior segment's structure and anatomical variations are essential for preventing glaucoma-related blindness.

Anterior segment optical coherence tomography (AS-OCT) is a rapid, high-resolution, non-invasive imaging modality. It provides detailed visualization of the anterior chamber and angle structures while measuring critical biological parameters. AS-OCT is essential for assessing the openness, narrowness, or closure of the anterior chamber angle and offers three-dimensional structural imaging and functional parameters. Due to its high sensitivity and accuracy, AS-OCT has become an essential tool for evaluating glaucoma severity, treatment efficacy, and prognosis (*Ang et al., 2018*; *Fujimoto, Swanson & Huang, 2023*). The rapid advancements in AS-OCT technology have spurred significant interest in glaucoma research. Despite many studies on AS-OCT advancements and clinical applications, systematic reviews and meta-analyses on specific aspects of glaucoma are still limited. With the exponential growth in research literature, employing novel methodologies is crucial for reviewing and analyzing research hotspots and trends in the field of glaucoma AS-OCT.

Bibliometric analysis is a quantitative research method. It uses mathematical and statistical techniques to extract, organize, and summarize data from literature. This method provides a comprehensive overview of development pathways, research hotspots, and trends within a field through knowledge mapping. It is becoming increasingly prevalent in ophthalmology (*Feng et al., 2023a*; *Wang et al., 2022*). Although there is extensive literature on various aspects of glaucoma AS-OCT, detailed analyses of development trends and research hotspots are still limited. Therefore, this study aims to conduct a bibliometric analysis of glaucoma AS-OCT research literature. The goal is to elucidating the trends and evolutionary trajectories in this rapidly evolving field.

## MATERIALS AND METHODS

### Data source and search strategy

This research uses the Web of Science (Core Collection) database as the source of bibliometric data. Web of Science encompasses a variety of research journals and provides various bibliometric indicators (such as titles, institutions, countries/regions, publication years, categories, and keywords). To ensure the accuracy and authority of the retrieved data, the indexes chosen are SCI-EXPANDED index.

Given the development history of the anterior segment optical coherence tomography in glaucoma, our search strategy was designed to capture relevant literature by employing the following search terms: TS = ("anterior segment OCT" OR "anterior segment optical coherence tomography" OR "AS-OCT" OR "anterior segment imaging") AND TS = ("glaucoma" OR "ocular hypertension" OR "intraocular pressure" OR "chamber angle").

The search strategy includes limiting databases, search terms, language, document type, and publication date. The search covered the period from January 2004 to December 2023, with the latest update on January 1, 2024. We included only articles and reviews. The documents were exported in "plain text" format, including "full records and references". Key data extracted comprised country or region, organizations, journals, authors, titles, abstracts, keywords, and references. After removing duplicates, we obtained 999 journal articles. Following a rigorous screening process, 931 valid articles were included in our analysis. The detailed search and analysis processes are illustrated in Fig. 1.

### Data analysis

In this research, we used several tools for analysis and visualization. These include R language bibliometrics (developed by Massimo Aria and Corrado Cuccurullo), VOSviewer (version 1.6.20, from the Centre for Science and Technology Studies at Leiden University, The Netherlands), and CiteSpace (version 6.2.R2, from Drexel University, PA, United States). Statistical analysis and bibliometric visualizations were performed using these tools. To ensure accuracy and relevance, two professional researchers specializing in ophthalmology independently conducted the data screening and analysis (*Wu et al., 2024*).

VOSviewer is a tool for creating and visualizing bibliometric maps. It performs co-occurrence analysis by examining the similarity of research topics. The tool extracts high-frequency keywords and organizes them into clusters to identify research hotspots. In network visualizations, each keyword is represented as a node. The size of each node is proportional to its frequency of occurrence. Links between nodes illustrate co-occurrence relationships (*Feng et al., 2023b*).

Using VOSviewer, we visualized collaboration maps among countries/regions, institutions, journals, and authors. The size of each item reflects its significance. Color coding represents different clusters. The total link strength indicates the extent of collaboration. Higher values signify stronger collaborative relationships.

CiteSpace generates visual bibliometric maps and node-link diagrams. It uses co-citation analysis, co-authorship networks, keyword clustering, and burst detection. This
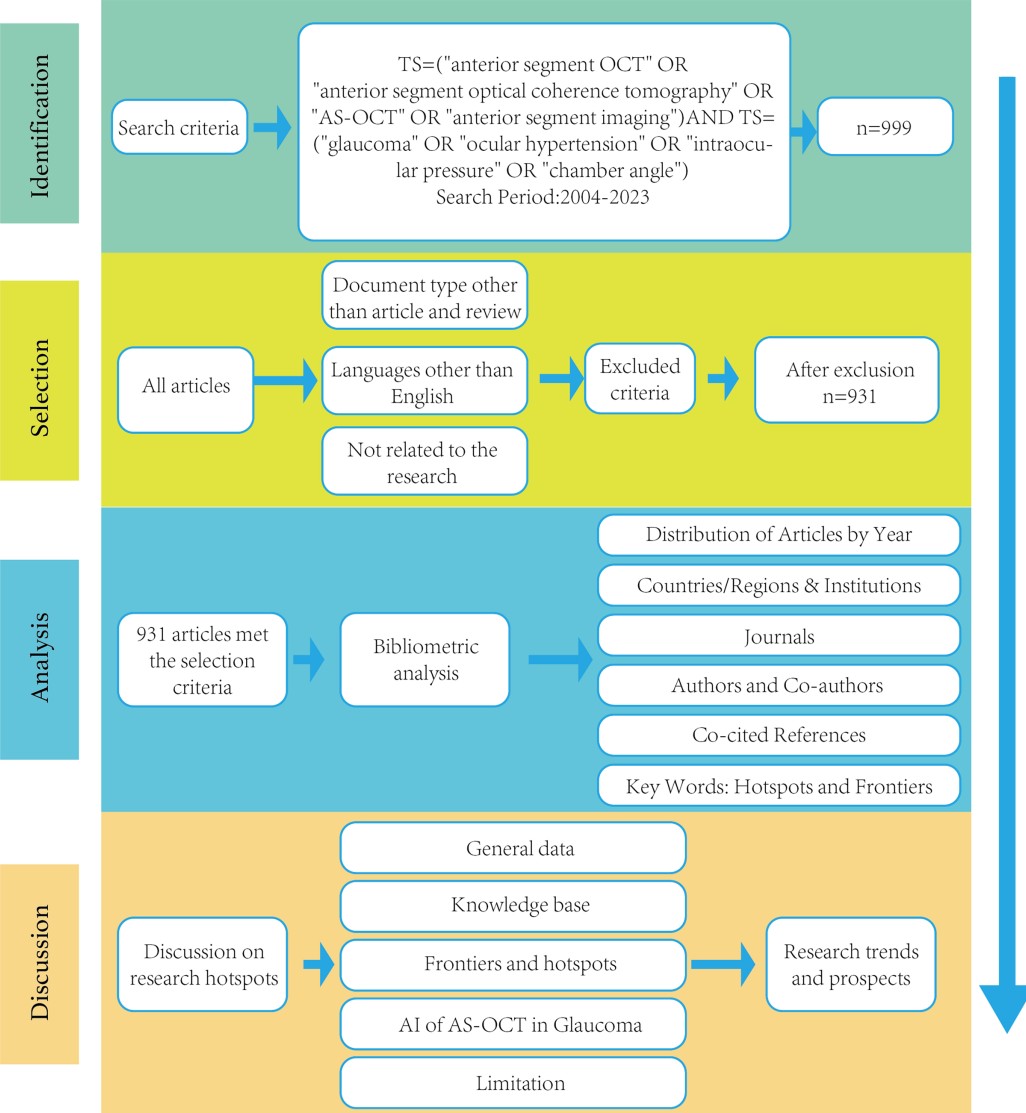

**Figure 1 Frame flow diagram for the detailed selection criteria and bibliometric analysis steps of applying AS-OCT to the study of glaucoma in the Web of Science Core Collection database.**

tool explores the knowledge structure, development trends, research hotspots, and emerging dynamics within the field (*Chen, 2004*).

The log-likelihood ratio algorithm is used for automatic labeling of co-citation clusters in CiteSpace. The tool generates key structural metrics, including centrality, modularity (Q score), and silhouette score (S score). Centrality measures how frequently a node connects with other nodes through the shortest paths. Nodes with high centrality are crucial for linking distinct clusters. The Q score, ranging from 0 to +1, assesses clustering within the co-citation network. Scores above 0.3 indicate significant clustering, with higher values reflecting better structure. The S score, ranging from −1 to +1, evaluates consistency within data clusters. Scores >0.3 indicate homogeneity, >0.5 indicate coherence, and >0.7

indicate high reliability. Burst detection identifies rapid increases in interest toward specific nodes. The temporal perspective of co-citation analysis shows the evolution of research topics. Each horizontal line represents a cluster, with research nodes depicted as "tree rings".

## RESULTS

This analysis includes 931 articles. They were authored by 3,458 individuals from 59 countries and 969 institutions. The articles appear in 116 journals and reference 14,234 documents across 1,980 journals.

### Annual distributions of publications

Our search identified 931 articles published from 2004 to 2023. Figure 2 summarizes the number of studies on AS-OCT in glaucoma over time. The volume of publications has steadily increased, with a notable surge after 2012. Since 2015, annual publications have consistently exceeded 60. This trend highlights growing scholarly interest and establishes AS-OCT as a prominent focus in glaucoma research.

### Top ten productive countries

This study analyzed publication volumes from 59 countries to identify those with the most significant contributions to this research field. Using VOSviewer, we visualized countries with five or more publications, as shown in Fig. 3. In this figure, node size represents publication volume. Line thickness indicates collaboration frequency between countries. Colors differentiate clusters.

The figure shows an uneven distribution of publishing countries. There is a notable concentration of articles from a few leading nations. The top 10 countries contribute to over half of the total publications in AS-OCT glaucoma research. The United States, China, and Singapore have more extensive collaborative efforts with other nations.

Further analysis identified the top 10 countries by publication volume, detailed in Table 1. Scholars from the United States lead with 288 research articles. They also have the highest number of citations and the highest intermediary centrality score. China follows with 231 publications, 5,144 citations, and an intermediary centrality score of 0.32.

### Top ten organizations

This study analyzed the publication output of 969 institutions to identify significant contributors to this research field. Co-authorship analysis using VOSviewer visualizes the institutional network in AS-OCT glaucoma research, as shown in Fig. 4. In the figure, node size represents the volume of published articles. Link strength indicates the intensity of collaboration among institutions. The distribution of institutions reveals a marked clustering effect, highlighting that academic research is primarily concentrated in a few leading countries.

Table 2 lists the top 10 institutions by publication volume in AS-OCT glaucoma research, contributing 56.2% of the documents (523 articles in total). The NATIONAL UNIVERSITY OF SINGAPORE leads all institutions with 93 publications, closely followed by SUN YAT SEN UNIVERSITY with 92. The SINGAPORE NATIONAL EYE CENTRE

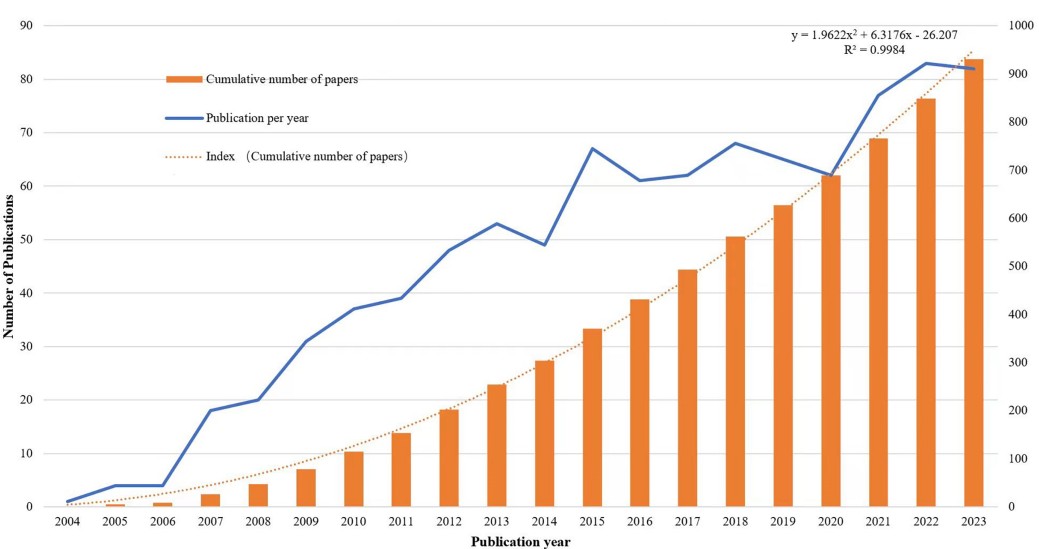

**Figure 2** **Trends in the number of publications on the applying AS-OCT to the study of glaucoma from 2004 to 2023.**

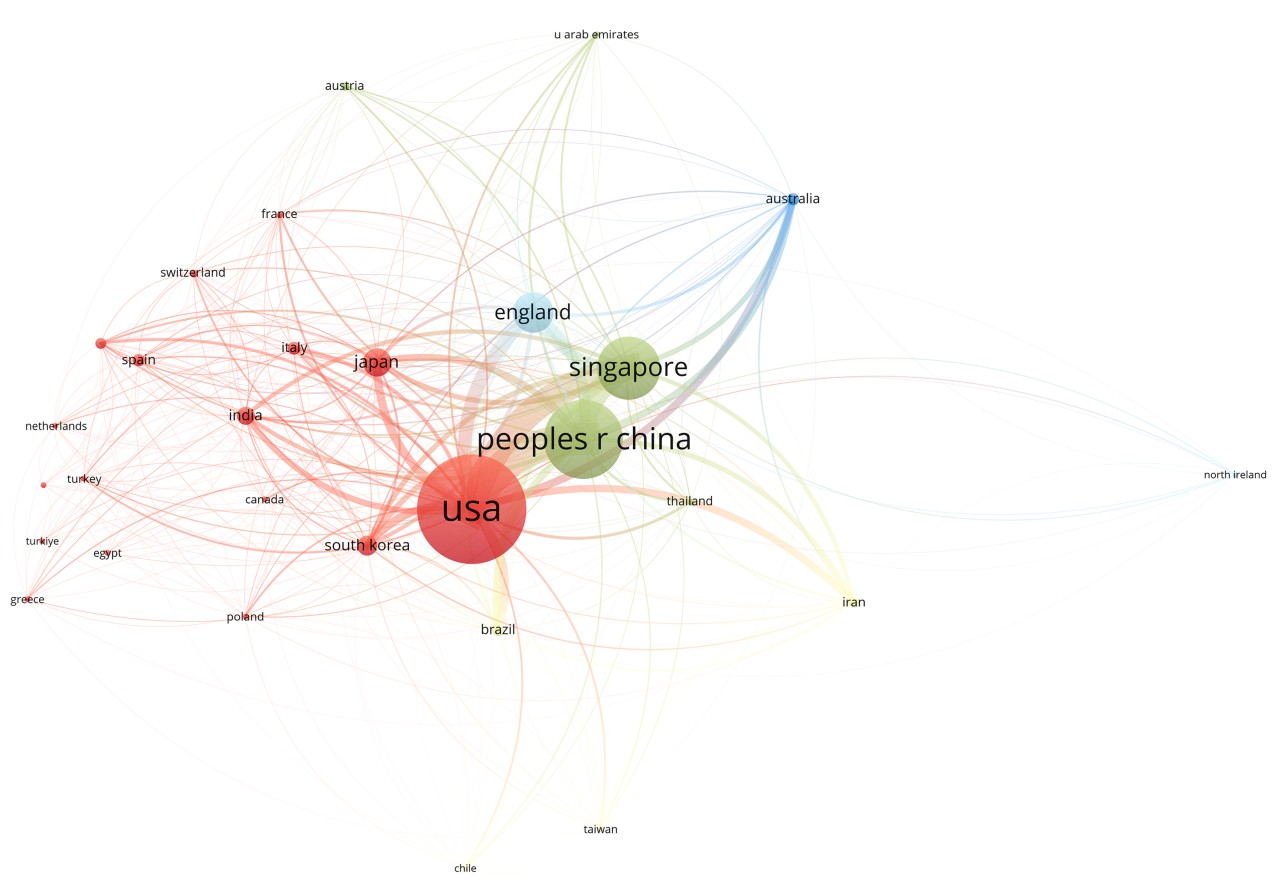

**Figure 3** **Distribution of main countries from 2004 to 2023 in AS-OCT to the study of glaucoma.**

**Table 1 The top 10 countries or regions with publications on the applying of AS-OCT in glaucoma from 2004 to 2023.**

| Rank | Countries/regions | Count | Citations | Centrality |
| --- | --- | --- | --- | --- |
| 1 | USA | 288 | 9,202 | 0.6 |
| 2 | China | 231 | 5,144 | 0.32 |
| 3 | Singapore | 124 | 4,743 | 0.05 |
| 4 | Japan | 98 | 1,332 | 0.07 |
| 5 | england | 75 | 3,140 | 0.26 |
| 6 | South Korea | 72 | 1,006 | 0.06 |
| 7 | India | 60 | 848 | 0 |
| 8 | Australia | 37 | 502 | 0.04 |
| 9 | Italy | 36 | 656 | 0.01 |
| 10 | Germany | 33 | 462 | 0.01 |

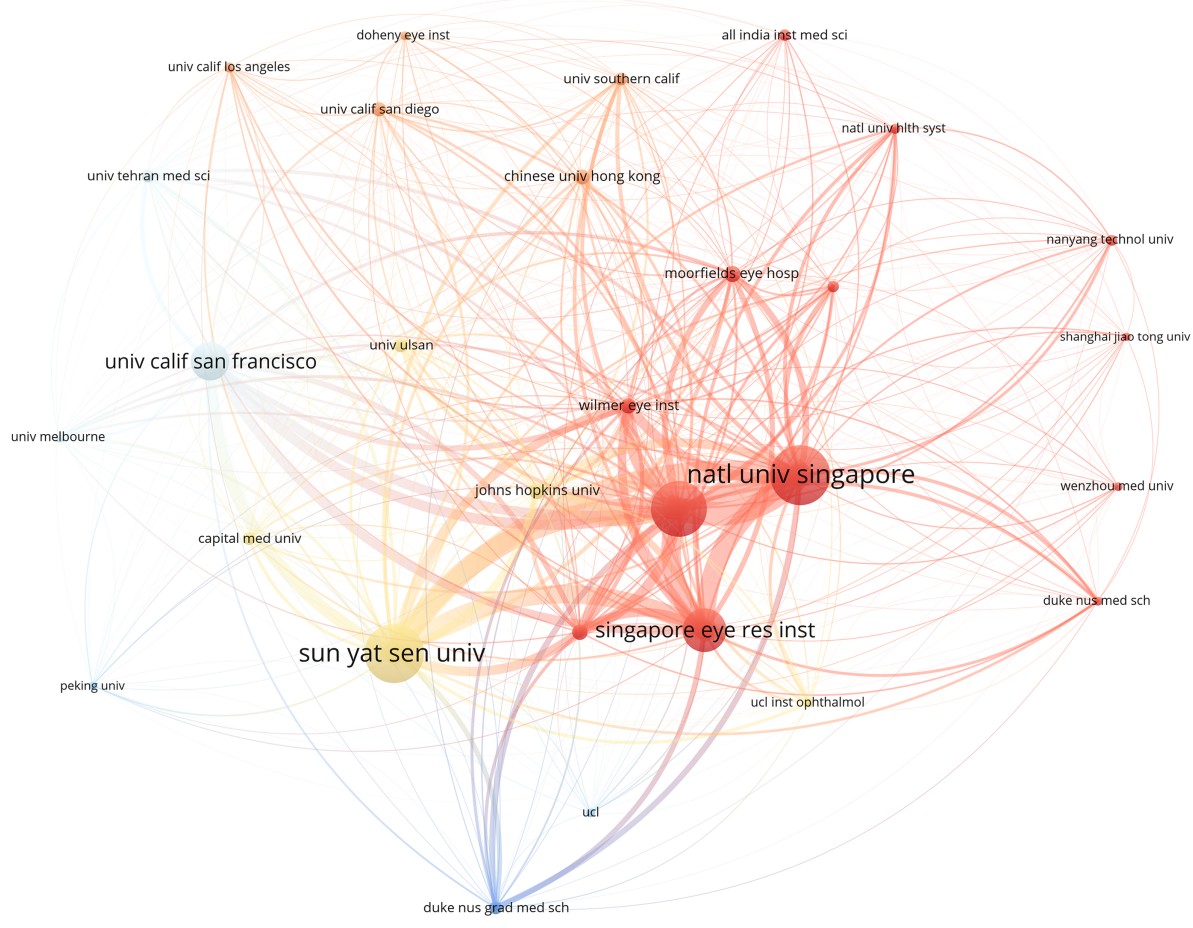

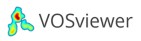 VOSviewer

**Figure 4 Cooperation network map of main research organizations that contributed to publications on the applying of AS-OCT in glaucoma from 2004 to 2023.**

**Table 2 The top 10 countries or regions with publications on the applying of AS-OCT in glaucoma from 2004 to 2023.**

| Rank | Affiliations (country) | Count | Citations | Centrality |
|---|---|---|---|---|
| 1 | NATIONAL UNIVERSITY OF SINGAPORE (SINGAPORE) | 93 | 3,536 | 0.04 |
| 2 | SUN YAT SEN UNIVERSITY (CHINA) | 92 | 2,540 | 0.07 |
| 3 | SINGAPORE NATIONAL EYE CENTRE (SINGAPORE) | 87 | 3,787 | 0.04 |
| 4 | SINGAPORE EYE RESEARCH INSTITUTE (SINGAPORE) | 68 | 2,550 | 0.07 |
| 5 | UNIVERSITY OF CALIFORNIA SAN FRANCISCO (USA) | 60 | 1,378 | 0.2 |
| 6 | JOHNS HOPKINS UNIVERSITY (USA) | 26 | 1,572 | 0.02 |
| 7 | MOORFIELDS EYE HOSPITAL (ENGLAND) | 26 | 1,695 | 0.01 |
| 8 | JOHNS HOPKINS BLOOMBERG SCHOOL OF PUBLIC HEALTH (USA) | 24 | 1,774 | 0.02 |
| 9 | WILMER EYE INSTITUTE (USA) | 24 | 2,118 | 0.02 |
| 10 | THE CHINESE UNIVERSITY OF HONG KONG (CHINA) LSAN UNIVERSITY (SOUTH KOREA) | 23 | 918 | 0.03 |

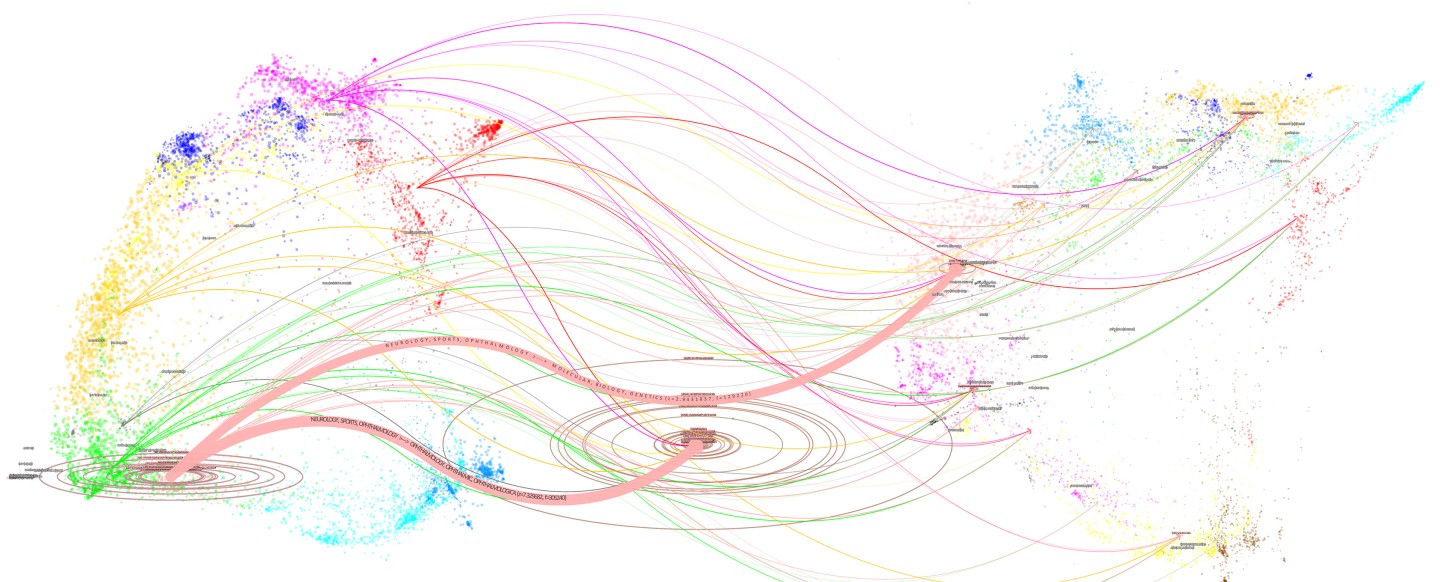

**Figure 5 Dual map overlay of journals that contributed to publications on the use of AS-OCT in glaucoma from 2004 to 2023.**

has the highest number of citations, whereas the UNIVERSITY OF CALIFORNIA SAN FRANCISCO holds the highest intermediary centrality.

## Distribution of journals

Citation relationships among academic journals illustrate knowledge exchange across diverse fields. The dual map of journals (Fig. 5) displays citing journals on the left and cited journals on the right, with colored lines indicating interdisciplinary citation relationships Journals in ophthalmology, mathematics, and clinical disciplines frequently cite research from bioengineering, medicine, and molecular biology. Ellipses in the Fig. 6 represent journal publication volume. The length of an ellipse indicates the number of authors, while the width represents the number of articles. Over the past two decades, AS-OCT glaucoma

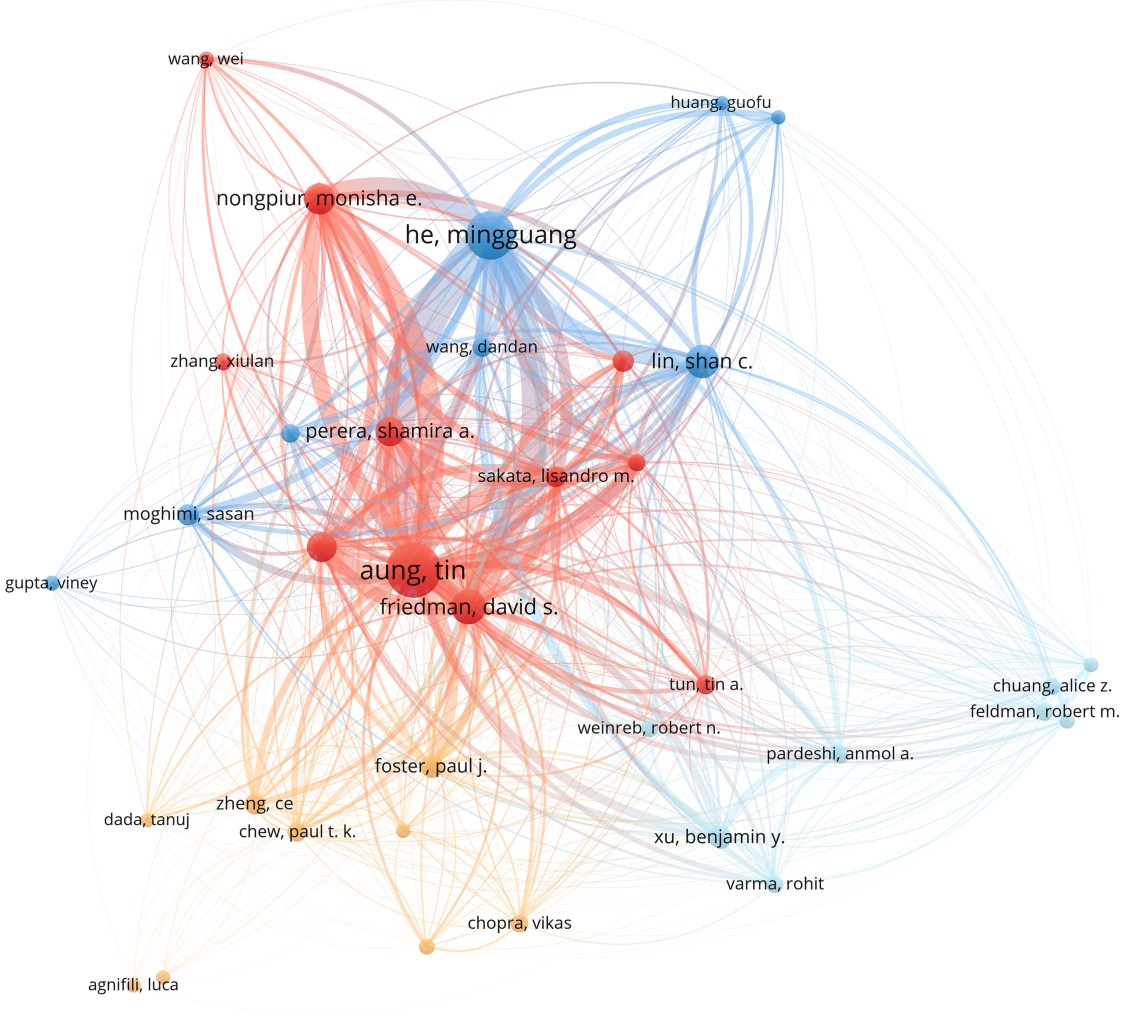

**Figure 6 Co-authorship network of productive authors that contributed to publications on the applying of AS-OCT in glaucoma from 2004 to 2023.**

research has primarily appeared in journals focused on ophthalmology, bioengineering medicine, and interdisciplinary fields. Some publications are also found in generalist journals. Node colors indicate discipline clusters, positions reflect the initial year of appearance, and lines show co-occurrence relationships among disciplines within clusters. Pink represents the most influential field.

Additionally, Table 3 lists the journals with the highest number of publications. Journals publishing over 40 articles include JOURNAL OF GLAUCOMA (86), INVESTIGATIVE OPHTHALMOLOGY & VISUAL SCIENCE (69), OPHTHALMOLOGY (46), and RITISH JOURNAL OF OPHTHALMOLOGY (44).

Analysis of citation data (see Supplemental File) reveals that PROGRESS IN RETINAL AND EYE RESEARCH is the most cited journal, with an average of 130.5 citations per article. This highlights the broad interest in its high-quality publications on glaucoma and

**Table 3 The top 10 main source journals with publications on the applying of AS-OCT in glaucoma from 2004 to 2023.**

| Rank | Journal | Country | Publications | Citations | Average citation/publication | Impact factor in 2023 | H-index |
|---|---|---|---|---|---|---|---|
| 1 | Journal of Glaucoma | USA | 86 | 1,301 | 15.1 | 2.29 | 20 |
| 2 | Investigative Ophthalmology & Visual Science | USA | 69 | 2,190 | 31.7 | 4.925 | 27 |
| 3 | Ophthalmology | USA | 46 | 3,443 | 74.8 | 14.277 | 30 |
| 4 | British Journal of Ophthalmology | England | 44 | 1,425 | 32.4 | 5.907 | 18 |
| 5 | American Journal of Ophthalmology | USA | 37 | 968 | 26.2 | 5.488 | 15 |
| 6 | BMC Ophthalmology | England | 33 | 126 | 3.9 | 2.086 | 6 |
| 7 | Journal of Cataract and Refractive Surgery | England | 32 | 979 | 30.6 | 3.528 | 15 |
| 8 | Cornea | USA | 30 | 482 | 16.1 | 3.152 | 12 |
| 9 | Graefes Archive for Clinical and Experimental Ophthalmology | Germany | 29 | 423 | 14.6 | 3.535 | 12 |
| 10 | Acta Ophthalmologica | Denmark | 27 | 589 | 21.8 | 3.988 | 13 |

**Table 4 The top 10 productive authors with publications on the applying of AS-OCT in glaucoma from 2004 to 2023.**

| Rank | Author | Publications | Citations | Average citation/publication | H-index | G-index |
|---|---|---|---|---|---|---|
| 1 | Aung, Tin | 80 | 3,595 | 44.9 | 33 | 60 |
| 2 | He, Mingguang | 65 | 2,414 | 37.1 | 28 | 49 |
| 3 | Friedman, David S. | 39 | 2,814 | 72.2 | 29 | 41 |
| 4 | Lin, Shan C. | 36 | 810 | 22.5 | 17 | 26 |
| 5 | Nongpiur, Monisha E. | 32 | 1,308 | 40.9 | 20 | 36 |
| 6 | Baskaran, Mani | 30 | 1,460 | 48.7 | 20 | 29 |
| 7 | Perera, Shamira A. | 29 | 795 | 27.4 | 19 | 27 |
| 8 | Foster, Paul J. | 20 | 1,339 | 66.9 | 17 | 21 |
| 9 | Xu, Benjamin Y. | 20 | 328 | 16.4 | 9 | 16 |
| 10 | Sung, Kyung Rim | 18 | 409 | 22.7 | 13 | 18 |

AS-OCT. Most publications in this journal are review articles. They discuss recent developments in AS-OCT for glaucoma research and clinical practice, including anterior segment monitoring.

## Distribution of authors and co-authors

Approximately 3,458 authors have contributed to a total of 931 documents. Table 4 shows the top contributors. Aung Tin leads with 80 documents and 3,595 citations, averaging 44.9 citations per document. He Mingguang follows with 65 documents and 2,414 citations, averaging about 37.1 citations per document.

Figure 6 illustrates the network of collaboration among authors. Node size indicates the number of articles by an author, while lines denote collaborative ties, with thicker lines signifying greater density of collaboration.

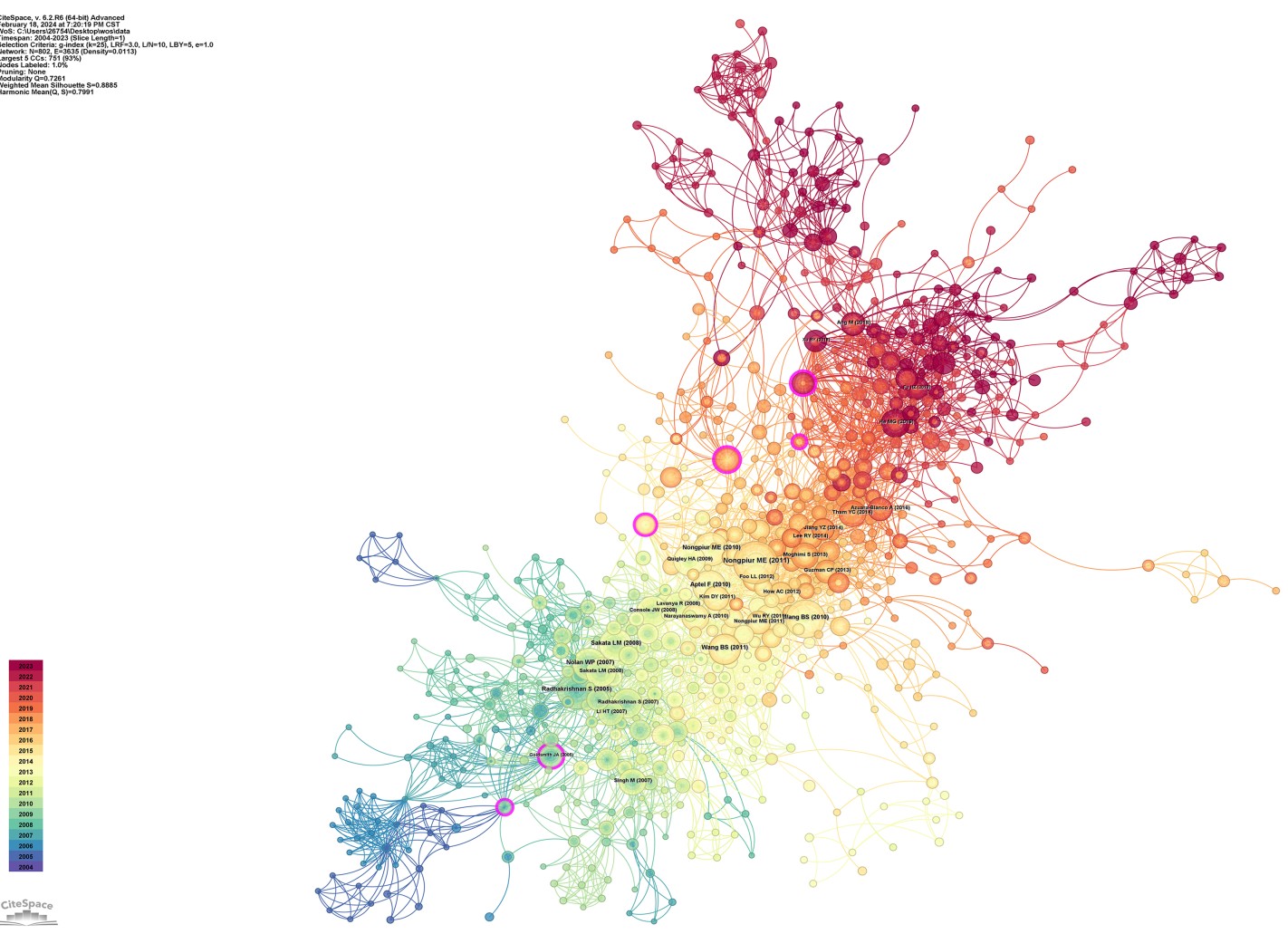

**Figure 7 The co-citation network map of references that applying of AS-OCT in glaucoma from 2004 to 2023.**

## Distribution of co-cited references

In our comprehensive investigation into the application of AS-OCT technology in glaucoma research, we identified 931 relevant articles that collectively cite 14,234 references, averaging 15.3 references per article. This indicates a substantial knowledge base in AS-OCT studies related to glaucoma.

Using CiteSpace's citation analysis tool, we constructed a network of cited references to assess scientific interconnections among the articles (Fig. 7). Key literature nodes in this network, linking two or more clusters, exhibit substantial intermediary centrality. These nodes act as crucial connectors between clusters, facilitating knowledge dissemination. They represent pivotal points in the discipline's knowledge base, significantly influencing its evolution and advancement.

The network clustering was configured with a g-index of K = 25 and a time slice of 1 year. The modularity Q score is 0.7261, which is above 0.5. This indicates that the network is effectively organized into loosely connected clusters. The weighted average

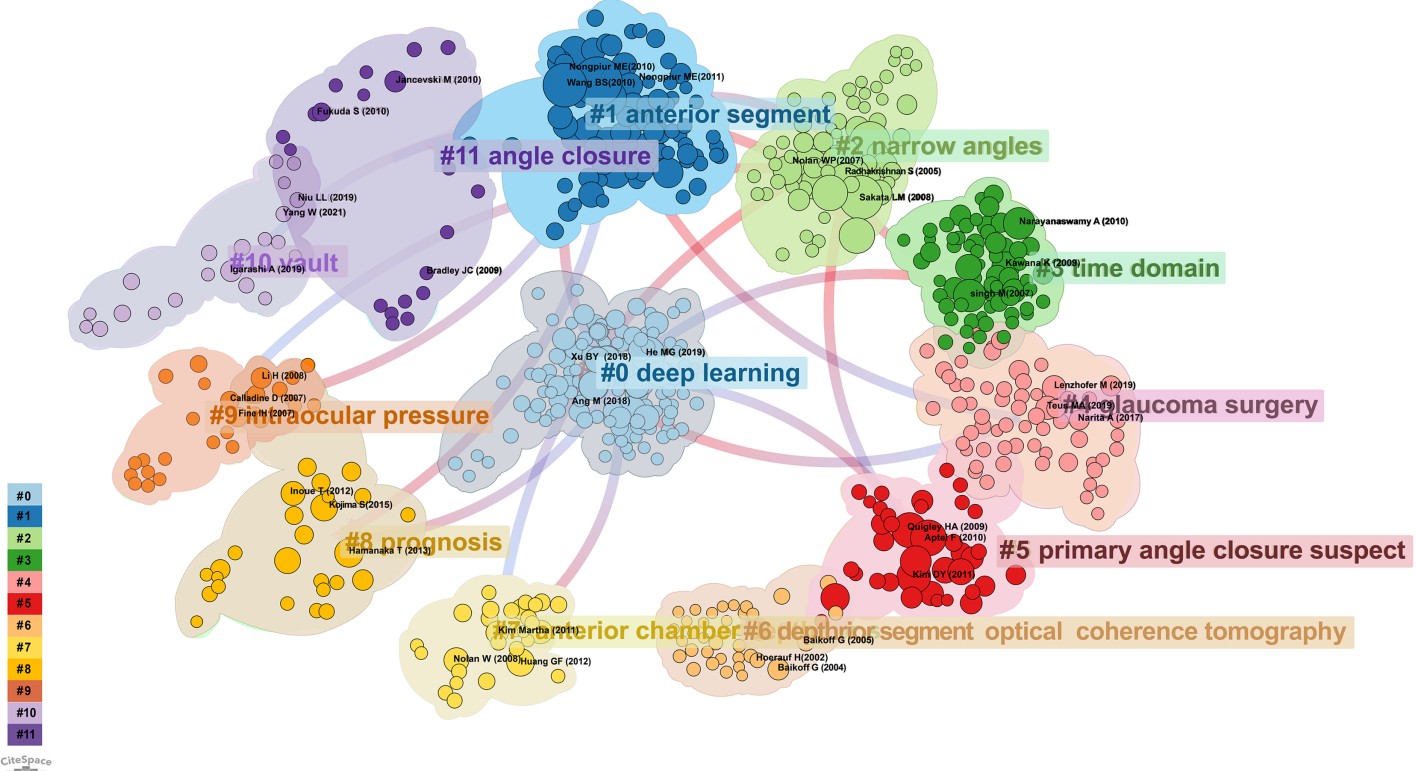

**Figure 8 The cluster map of co-cited references that applying of AS-OCT in glaucoma from 2004 to 2023.**

silhouette score is 0.8885, also above 0.5. This confirms the strong coherence of the clusters. These metrics suggest a well-defined modular structure within the network. "Anterior segment" and "deep learning" have emerged as the predominant research communities.

Following the citation clustering, Fig. 8 highlights the pivotal nodes within the clusters, delineating the knowledge base of this field of study. They are identified as: #0 deep learning (*Fu et al., 2019*), #1 anterior segment (*Moghimi et al., 2013*), #2 narrow angles (*Narayanaswamy et al., 2010*), #3 time domain, #4 glaucoma surgery (*Singh et al., 2007*), #5 primary angle closure suspect, #6 anterior segment optical coherence tomography, #7 anterior chamber depth, #8 prognosis (*Lavanya et al., 2008*), #9 intraocular pressure, #10 vault, and #11 angle closure.

Citation bursts were observed from 2004 to 2023 (Fig. 9). Notably, the study by *Radhakrishnan et al. (2005)* showed the most intense citation burst, with a strength of 25.7. The research by *Fu et al. (2019)* displayed the most recent burst, with a strength of 9.49. These significant bursts highlight critical developments and milestones in the application of AS-OCT within glaucoma research.

## Top 25 References with the Strongest Citation Bursts

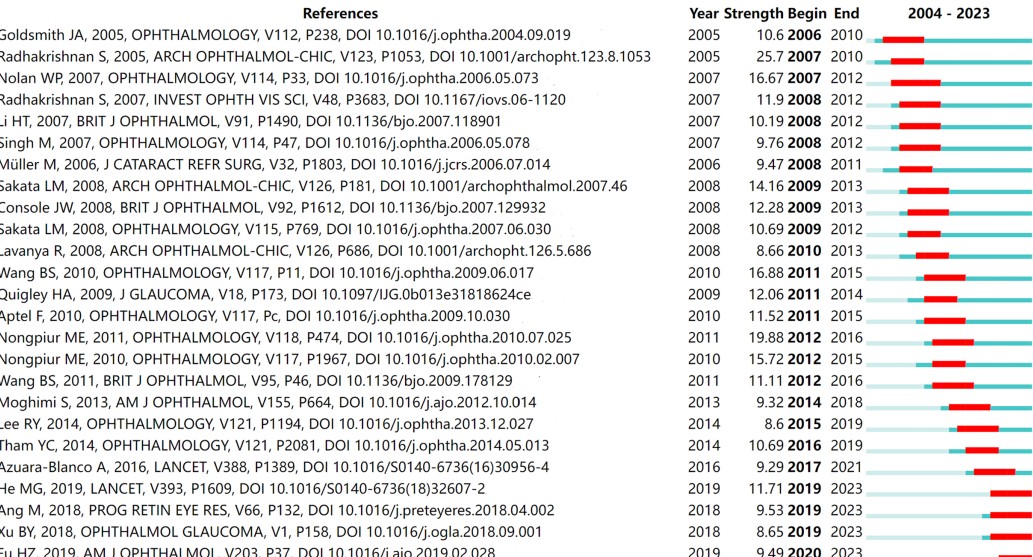

| References | Year | Strength | Begin | End | 2004 - 2023 |
|---|---|---|---|---|---|
| Goldsmith JA, 2005, OPHTHALMOLOGY, V112, P238, DOI 10.1016/j.ophtha.2004.09.019 | 2005 | 10.6 | **2006** | 2010 | |
| Radhakrishnan S, 2005, ARCH OPHTHALMOL-CHIC, V123, P1053, DOI 10.1001/archopht.123.8.1053 | 2005 | 25.7 | **2007** | 2010 | |
| Nolan WP, 2007, OPHTHALMOLOGY, V114, P33, DOI 10.1016/j.ophtha.2006.05.073 | 2007 | 16.67 | **2007** | 2012 | |
| Radhakrishnan S, 2007, INVEST OPHTH VIS SCI, V48, P3683, DOI 10.1167/iovs.06-1120 | 2007 | 11.9 | **2008** | 2012 | |
| Li HT, 2007, BRIT J OPHTHALMOL, V91, P1490, DOI 10.1136/bjo.2007.118901 | 2007 | 10.19 | **2008** | 2012 | |
| Singh M, 2007, OPHTHALMOLOGY, V114, P47, DOI 10.1016/j.ophtha.2006.05.078 | 2007 | 9.76 | **2008** | 2012 | |
| Müller M, 2006, J CATARACT REFR SURG, V32, P1803, DOI 10.1016/j.jcrs.2006.07.014 | 2006 | 9.47 | **2008** | 2011 | |
| Sakata LM, 2008, ARCH OPHTHALMOL-CHIC, V126, P181, DOI 10.1001/archophthalmol.2007.46 | 2008 | 14.16 | **2009** | 2013 | |
| Console JW, 2008, BRIT J OPHTHALMOL, V92, P1612, DOI 10.1136/bjo.2007.129932 | 2008 | 12.28 | **2009** | 2013 | |
| Sakata LM, 2008, OPHTHALMOLOGY, V115, P769, DOI 10.1016/j.ophtha.2007.06.030 | 2008 | 10.69 | **2009** | 2012 | |
| Lavanya R, 2008, ARCH OPHTHALMOL-CHIC, V126, P686, DOI 10.1001/archopht.126.5.686 | 2008 | 8.66 | **2010** | 2013 | |
| Wang BS, 2010, OPHTHALMOLOGY, V117, P11, DOI 10.1016/j.ophtha.2009.06.017 | 2010 | 16.88 | **2011** | 2015 | |
| Quigley HA, 2009, J GLAUCOMA, V18, P173, DOI 10.1097/IJG.0b013e31818624cd | 2009 | 12.06 | **2011** | 2014 | |
| Aptel F, 2010, OPHTHALMOLOGY, V117, Pc, DOI 10.1016/j.ophtha.2009.10.030 | 2010 | 11.52 | **2011** | 2015 | |
| Nongpiur ME, 2011, OPHTHALMOLOGY, V118, P474, DOI 10.1016/j.ophtha.2010.07.025 | 2011 | 19.88 | **2012** | 2016 | |
| Nongpiur ME, 2010, OPHTHALMOLOGY, V117, P1967, DOI 10.1016/j.ophtha.2010.02.007 | 2010 | 15.72 | **2012** | 2015 | |
| Wang BS, 2011, BRIT J OPHTHALMOL, V95, P46, DOI 10.1136/bjo.2009.178129 | 2011 | 11.11 | **2012** | 2016 | |
| Moghimi S, 2013, AM J OPHTHALMOL, V155, P664, DOI 10.1016/j.ajo.2012.10.014 | 2013 | 9.32 | **2014** | 2018 | |
| Lee RY, 2014, OPHTHALMOLOGY, V121, P1194, DOI 10.1016/j.ophtha.2013.12.027 | 2014 | 8.6 | **2015** | 2019 | |
| Tham YC, 2014, OPHTHALMOLOGY, V121, P2081, DOI 10.1016/j.ophtha.2014.05.013 | 2014 | 10.69 | **2016** | 2019 | |
| Azuara-Blanco A, 2016, LANCET, V388, P1389, DOI 10.1016/S0140-6736(16)30956-4 | 2016 | 9.29 | **2017** | 2021 | |
| He MG, 2019, LANCET, V393, P1609, DOI 10.1016/S0140-6736(18)32607-2 | 2019 | 11.71 | **2019** | 2023 | |
| Ang M, 2018, PROG RETIN EYE RES, V66, P132, DOI 10.1016/j.preteyeres.2018.04.002 | 2018 | 9.53 | **2019** | 2023 | |
| Xu BY, 2018, OPHTHALMOL GLAUCOMA, V1, P158, DOI 10.1016/j.ogla.2018.09.001 | 2018 | 8.65 | **2019** | 2023 | |
| Fu HZ, 2019, AM J OPHTHALMOL, V203, P37, DOI 10.1016/j.ajo.2019.02.028 | 2019 | 9.49 | **2020** | 2023 | |

**Figure 9** The top 25 references with the strongest citation bursts on AS-OCT in glaucoma from 2004 to 2023. The references: Goldsmith et al. (2005), Radhakrishnan et al. (2005), Nolan et al. (2007), Radhakrishnan et al. (2007), Li et al. (2007), Singh et al. (2007), Müller et al. (2006), Sakata et al. (2008b), Console et al. (2008), Sakata et al. (2008a), Lavanya et al. (2008), Wang et al. (2010), Quigley et al. (2009), Aptel & Denis (2010), Nongpiur et al. (2011, 2010), Wang et al. (2011), Moghimi et al. (2013), Lee et al. (2014), Tham et al. (2014), Azuara-Blanco et al. (2016), He et al. (2019), Ang et al. (2018), Xu et al. (2018), Fu et al. (2019).

## Distribution of key words: hotspots and frontiers

Keywords encapsulate the core content of articles. Analyzing the co-occurrence of these keywords reveals research hotspots and emerging trends. From 931 articles, we extracted 2,441 keywords. We conducted a keyword co-occurrence network analysis to explore the evolution of these keywords over time. This provided insights into AS-OCT's role in diagnosing and treating glaucoma and tracked shifts in research focus. The CiteSpace configuration was set to "Year Per Slice" = 1, "Top N%" = 10.0%, and "Most Duration" = 1.

Figure 10 illustrates the temporal distribution of keywords, showing shifts in research focus over time. Before 2008, AS-OCT studies mainly focused on reproducibility, comparison, and anterior segment swept-source optical coherence tomography. From 2008 to 2013, the emphasis shifted to lens vault, iris parameters, and depth. Between 2013 and 2018, research attention moved towards management, surgery, 3D imaging, and angiography. From 2018 to 2023, machine learning, deep learning, and predictors became prominent areas of interest.

The strongest citation bursts highlight terms with significant increases in citation frequency over specific periods. This indicates frontier developments in AS-OCT for glaucoma research. Figure 11 shows the top 25 keywords with the most pronounced citation bursts. The timeline is represented by a blue line, with red segments indicating the duration of each burst and endpoints marking its start and end. "Ultrasound biomicroscopy" shows the most intense citation burst. "Biometry" had the longest

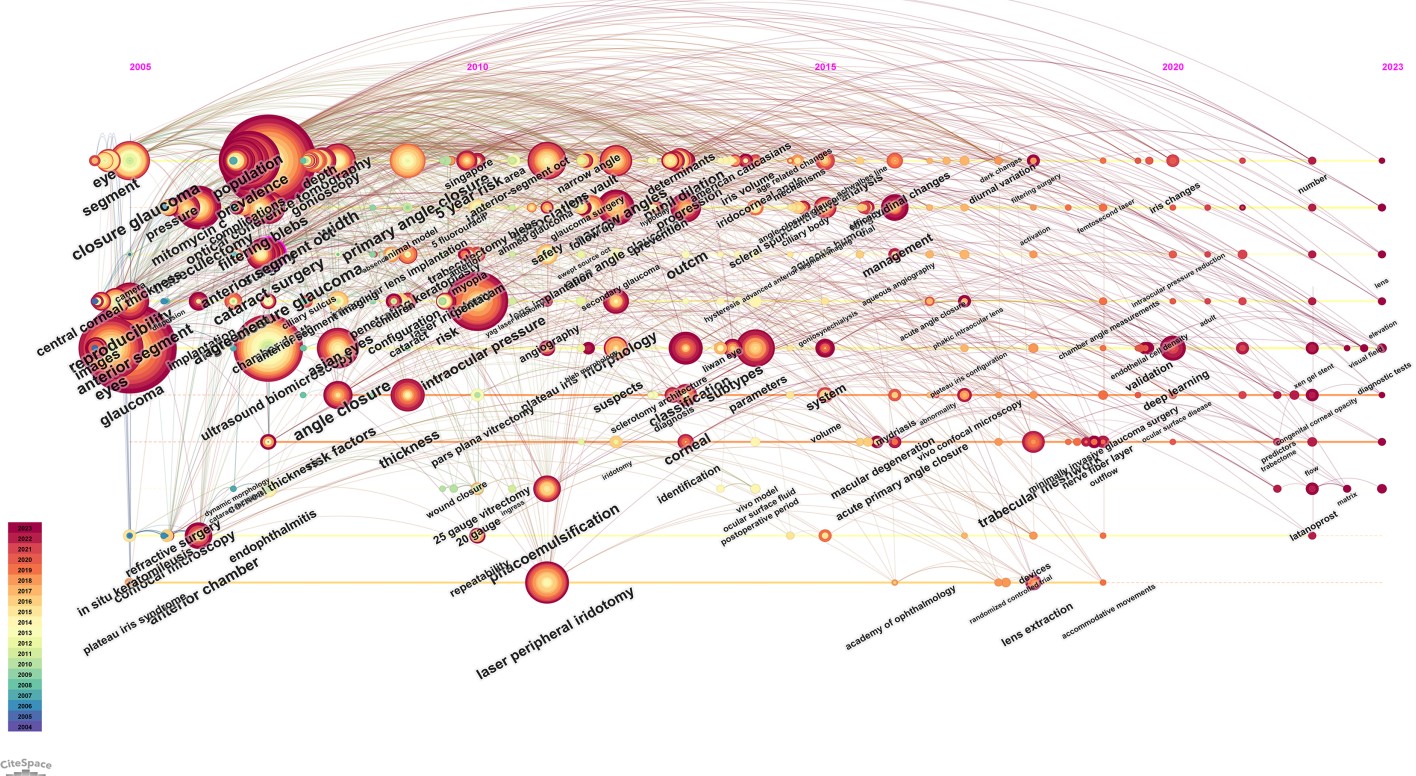

**Figure 10  The timeline visualization of keywords that applying of AS-OCT in glaucoma from 2004 to 2023.**

duration of prominence (2007–2015). Keywords like "classification" and "deep learning" gained traction from 2019 to 2023. Clinically relevant terms include "closure glaucoma," "depth," "suspects," and "ultrasound biomicroscopy." Technically significant terms include "deep learning," "classification," "progression," and "management".

## DISCUSSION

Bibliometric analysis is an essential method for identifying research hotspots and forecasting future trends. It converts literature relationships into scientific knowledge maps. These maps are widely regarded as effective tools for extracting valuable insights from complex literature networks. This study performed a bibliometric analysis on AS-OCT in glaucoma research, sourced from Web of Science, revealing key foundational details, research hotspots, and emerging trends.

### General data

This study utilized the Web of Science Core Collection SCI-EXPANDED to retrieve 931 articles related to glaucoma AS-OCT. These articles were published between 2004 and 2023. The volume of academic literature is a key indicator of the development trajectory in

## Top 25 Keywords with the Strongest Citation Bursts

| Keywords | Year | Strength | Begin | End | 2004 - 2023 |
|---|---|---|---|---|---|
| ultrasound biomicroscopy | 2007 | 14.82 | **2007** | 2011 | |
| biometry | 2007 | 4.28 | **2007** | 2015 | |
| angle closure glaucoma | 2007 | 3.19 | **2007** | 2013 | |
| closure glaucoma | 2005 | 7.14 | **2008** | 2013 | |
| depth | 2008 | 6.14 | **2008** | 2013 | |
| chamber | 2008 | 5.26 | **2008** | 2013 | |
| reproducibility | 2005 | 4.29 | **2008** | 2010 | |
| asian eyes | 2008 | 3.92 | **2008** | 2011 | |
| people | 2008 | 3.44 | **2008** | 2010 | |
| segment | 2005 | 4.33 | **2010** | 2015 | |
| lens implantation | 2011 | 4.09 | **2011** | 2013 | |
| primary angle closure | 2009 | 4.02 | **2012** | 2016 | |
| determinants | 2013 | 4.43 | **2013** | 2015 | |
| narrow angles | 2012 | 3.44 | **2013** | 2014 | |
| lens vault | 2012 | 3.4 | **2014** | 2017 | |
| phacoemulsification | 2011 | 3.3 | **2014** | 2016 | |
| open angle glaucoma | 2012 | 4.13 | **2015** | 2017 | |
| suspects | 2012 | 4.43 | **2016** | 2018 | |
| longitudinal changes | 2016 | 3.19 | **2016** | 2020 | |
| management | 2016 | 3.62 | **2017** | 2023 | |
| progression | 2013 | 3.38 | **2017** | 2020 | |
| trabecular meshwork | 2018 | 4.8 | **2018** | 2023 | |
| intraocular pressure | 2010 | 4.79 | **2018** | 2019 | |
| classification | 2013 | 4.65 | **2019** | 2023 | |
| deep learning | 2020 | 6.64 | **2020** | 2023 | |

**Figure 11 Keywords with the strongest citation bursts for publications on the use of AS-OCT in glaucoma from 2004 to 2023.**

this research field. The trend in article distribution illustrates the evolution of glaucoma AS-OCT technology. It shows the progression from its initial stages to more advanced applications. This progression includes rapid advances in clinical research and significant expansion in the artificial intelligence (AI) era. These developments highlight the substantial impact of glaucoma AS-OCT on advancements in medical systems.

In the field of ophthalmic AS-OCT research, the United States, China, and Singapore are the leading contributors. The United States stands out for both the quantity and quality of its research. China is noted for its rapid growth, while Singapore has made significant contributions. Additionally, the United States serves as a central hub for international collaborations. It works closely with China, Singapore, Japan, and the UK. This underscores that geographical distance has little impact on fostering international partnerships.

The NATIONAL UNIVERSITY OF SINGAPORE, SUN YAT SEN UNIVERSITY, SINGAPORE NATIONAL EYE CENTRE, SINGAPORE EYE RESEARCH INSTITUTE, and the UNIVERSITY OF CALIFORNIA SAN FRANCISCO stand out for their active and impactful roles in international collaborations.

To evaluate journal impact, we used the Impact Factor (IF) from the Journal Citation Reports' Science Citation Index. This index was developed by the Institute for Scientific Information in Philadelphia. The IF measures how often journals are cited in the SCI database over a 2-year period, relative to the number of articles they published during that time. Although the IF has limitations, it is widely regarded as a measure of journal prestige (*Garfield, 2006*). Our findings reveal that highly cited articles are published in top journals such as OPHTHALMOLOGY and PROGRESS IN RETINAL AND EYE RESEARCH. OPHTHALMOLOGY specializes in clinical ophthalmology, while PROGRESS IN RETINAL AND EYE RESEARCH focuses on ophthalmological reviews. Both are considered high-impact journals. Consequently, articles in high-impact journals generally receive more citations. Furthermore, an analysis of journal distribution shows that glaucoma AS-OCT research spans a broad range of journals. These include those in clinical ophthalmology, bioengineering, technological development, application, and artificial intelligence.

The H-index was employed to quantify researchers' scientific output. It serves as a common bibliometric measure for assessing scholarly achievements. However, the H-index has limitations. It may undervalue researchers who prioritize quality over quantity and tends to favor those with longer careers. To address these shortcomings, the G-index was introduced. The G-index provides a more comprehensive assessment of an article's overall citation impact. Notable researchers, such as Aung Tin, He Mingguang, and David S. Friedman are highlighted for their significant contributions and collaborative efforts (*Hirsch, 2005*).

## Knowledge base

Prior research has investigated factors influencing citation rates. Key determinants include article quality, journal impact factor, number of authors, visibility, and international collaboration. Analysis of co-authorship and citations reveals authors' contributions and impact. This provides insights into research collaborations within the AS-OCT imaging domain. Citation analysis also highlights prevalent themes in high-quality scholarly outputs.

In clinical ophthalmology, accurately understanding spatial relationships among eye structures is essential. Compared to ultrasound biomicroscopy (UBM) and gonioscopy, AS-OCT offers significant advantages. It provides rapid and non-invasive imaging of internal eye structures with quantitative measurement capabilities (*Mirzayev et al., 2023*; *Radhakrishnan et al., 2005*). In 1994, *Izatt et al. (1994)* first utilized 830 μm wavelength light for anterior segment OCT imaging. Later, Huang and Izatt introduced modern AS-OCT in 2001. They used 1,310 nm wavelength light and a scanning speed of 4,000 A per second, which improved axial resolution (*Goldsmith et al., 2005*). Time-domain AS-OCT has since evolved into Fourier-domain OCT, including swept-source and spectral-domain configurations. These advancements have led to faster scans, higher axial resolutions, and superior signal-to-noise ratios. As a result, image quality and
reproducibility have significantly improved. This progress has enriched anterior segment imaging and facilitated both qualitative and quantitative evaluations of anterior segment structures in glaucoma (*Moghimi et al., 2013*).

Previous studies have identified several anterior chamber measurements associated with gonioscopic angle closure. These measurements include anterior chamber width (AC width), lens vault (LV), iris curvature (IC), and iris thickness (IT). Notably, these anatomical risk factors can explain more than 80% of the variation in angle width (*Nongpiur et al., 2017*, *2013b*). Other key parameters studied are the angle opening distance and trabeculo-iris space area at 500 μm from the scleral spur (AOD500, TISA-500). Additionally, the anterior chamber angle, lens thickness, anterior chamber depth (ACD), and lens position are also important.

Among these measurements, a larger lens vault indicates that increased lens thickness contributes to crowding in the anterior chamber. *Nongpiur et al.*'s *(2011)* study on Chinese patients with angle-closure glaucoma identified a significant association between lens vault and angle closure. Specifically, when eyes were categorized into quartiles based on lens vault, the risk of angle closure in the quartile with the largest lens vault was 48 times higher than in the quartile with the smallest lens vaults. This finding highlights the independence of lens vault as a risk factor, separate from other angle-closure risk factors such as age, gender, and biological traits of the eye, including anterior chamber depth, lens thickness, and position. Similar research involving Japanese patients supported these results. It showed an odds ratio of 24.2 for angle closure when comparing the lowest to the highest quartiles of lens vault (*Ozaki et al., 2012*).

AS-OCT is instrumental in evaluating filtering bleb morphology, implant positioning, angle openness, and ciliary body block following anti-glaucoma surgery. It aids in assessing postoperative outcomes and complications. *Singh et al. (2007)* pioneered the use of AS-OCT for analyzing filtering blebs after trabeculectomy. They assessed parameters such as bleb height, thickness, intra-wall cavities or microcystic structures, scleral flap, its attachment to the underlying scleral tissue, and the condition of the internal ostium. Their study found that AS-OCT and slit-lamp examination provided consistent classifications of bleb height (*Singh et al., 2007*). Additionally, lens enlargement can lead to pupillary block, a key mechanism in primary angle closure disease (PACD). During cataract surgery, AS-OCT reveals angle widening and Schlemm's canal expansion. These findings are associated with decreased intraocular pressure (*Gong et al., 2024*; *Zhao et al., 2016*). Consequently, cataract surgery is considered a primary surgical intervention for relieving angle closure and reducing intraocular pressure in PACG patients (*Azuara-Blanco et al., 2016*).

In conclusion, research efforts predominantly focus on utilizing AS-OCT to investigate glaucoma pathogenesis, conduct biological assessments, evaluate diagnostic effectiveness, and guide surgical treatments.

## Frontiers and hotspots

Keyword co-occurrence and emergent keyword analysis uncover hot trends within the research domain, reflecting the focal research themes. Deep learning and classification

represent the latest explosive keywords within this field. Currently, AS-OCT imaging is constrained to the manual evaluation of each biological parameter, necessitating ophthalmologists to identify specific anatomical structures manually. This is to assess the state of the anterior chamber angle and the lens's transparency, among others, rendering the measurement process time-consuming, subjective, and with low repeatability.

## Artificial intelligence of anterior segment optical coherence tomography in glaucoma

Leveraging AI and extensive imaging data collected from clinical and research contexts conserves time for physicians and patients. It also enhances diagnostic capabilities (*Beam et al., 2023*; *Chen et al., 2023*; *Garcia Marin et al., 2022*; *LeCun, Bengio & Hinton, 2015*; *Scheetz, He & van Wijngaarden, 2021*). AI technologies have been developed to identify significant eye diseases. These include diabetic retinopathy (*Abràmoff et al., 2018*), primary open-angle glaucoma (*Russakoff et al., 2020*), visual impairment (*Tham et al., 2021*), and health risks in fundus images (*Rim et al., 2021*).

In 2013, *Nongpiur et al. (2013b)* evaluated classification algorithms using AS-OCT measurements to detect gonioscopic angle closure. The study involved 2,047 participants aged 50 years and older. Gonioscopy and AS-OCT scans were analyzed with custom software. The assessment used 10-fold cross-validation and training-validation splits to evaluate six classification algorithms. The findings identified stepwise logistic regression as the most effective algorithm, achieving an accuracy rate exceeding 95% in detecting gonioscopic angle closure based on six specific AS-OCT parameters (*Console et al., 2008*; *Nongpiur et al., 2013b*).

*Baek et al. (2013)* developed a clustering machine learning algorithm specifically for glaucoma-related characteristics. This algorithm identified two distinct feature sets within an angle-closure cohort. It established a fundamental binary classification framework. This framework was used to explore the underlying mechanisms of angle closure incidence within the group (*Baek et al., 2013*).

Furthermore, *Xu et al. (2013)* proposed a novel machine learning methodology using a support vector machine linear classifier. This research involved extracting both visual and clinical features. Clinical attributes included the area of Schwalbe's line, while visual attributes encompassed the Histogram of Oriented Gradients (HOG) and Biologically Inspired Features (BIF). The study involved preprocessing the dataset for anterior chamber angle area detection and feature extraction for angle-closure glaucoma detection. When evaluated on a dataset of 2048 images, the results showed an AUC value of 0.90. A classification precision of 80% was achieved using only visual features (*Xu et al., 2013*).

*Niwas et al. (2016)* analyzed both unsupervised (L-score) and supervised (MRMR) feature selection algorithms to identify angle-closure glaucoma (ACG) mechanisms using AS-OCT images. The investigation revealed that the L-score approach, which includes redundant features, achieved a higher accuracy rate of 86.66%. This was achieved using a broader feature set (40 out of 84). In contrast, the MRMR strategy, which uses fewer redundant features, achieved an accuracy of 84.39% with a more limited set (10 out of 84). The study concludes that incorporating redundant features could enhance ACG detection.

This underscores the potential advantage of the L-score method in medical diagnostics, particularly for complex conditions like ACG (*Niwas et al., 2016*).

Simultaneously, *Ni Ni et al. (2014)* used a similar methodology but introduced different feature inputs to a fuzzy k-nearest neighbors (fkNN) classifier. This classification technique identifies new instances by comparing them with labeled instances in the training dataset. Their study, conducted on a dataset of 264 SS-OCT images, demonstrated high efficacy. The results showed an AUC of 0.98 and an accuracy of 99% (*Ni Ni et al., 2014*).

While machine learning approaches are limited by their dependence on human-selected features for model training, which may restrict their effectiveness and generalizability, it is important to note that algorithms utilizing clinical data for classifying angle-closure eye disease mechanisms have been developed. Algorithms based on image data have also demonstrated strong capabilities in detecting angle closure. This highlights the significant impact and potential of these technological advancements in the field (*Wang et al., 2024*).

Deep learning algorithms, trained directly on image data, generally outperform traditional machine learning algorithms in detecting PACD. They excel at accurately distinguishing between angle states—open, narrow, and closed. Additionally, these algorithms automatically quantify parameters. As a result, there has been increasing emphasis on deep learning approaches in recent years. These approaches provide cutting-edge performance that is comparable to the manual analysis performed by trained human observers (*Porporato et al., 2020*).

Among these, deep learning algorithms have demonstrated high sensitivity and specificity in detecting and classifying retinal lesions and related eye conditions. This represents a promising area of research. Google researchers developed a deep learning-based algorithm for detecting diabetic retinopathy and macular edema. This algorithm is noted for its high specificity and sensitivity (*Gulshan et al., 2016*). Subsequently, several researchers have advanced the use of deep learning for detecting diabetic retinopathy (DR) (*Gargeya & Leng, 2017*; *Quellec et al., 2017*). Zhang Kang's research group used deep learning algorithms trained on fundus OCT images to diagnose age-related macular degeneration and other blinding eye diseases (*Kermany et al., 2018*).

*Hao et al. (2019)* employed a multi-scale convolutional neural network for the detailed classification of closed, narrow, and open angles. This technique has potential value for guiding various stages of clinical treatment, achieving an AUC value of 0.9143 (*Hao et al., 2019*). Additionally, they introduced a multi-sequence deep network that utilizes AS-OCT images captured in both illuminated and non-illuminated conditions. This approach integrates temporal and spatial information and achieved an F1 score of 0.855 (*Hao et al., 2021*).

Furthermore, *Pham et al. (2021)* developed a deep convolutional neural network (DCNN) for accurate scleral spur localization. Their approach incorporates a segmentation technique enriched with additional information to address localization challenges. This represents a significant advancement towards establishing a reliable automated framework for the accurate quantification of AS-OCT scans, which is crucial for the diagnosis and management of angle-closure glaucoma (*Pham et al., 2021*).

AS-OCT scans can detect the presence of angle closure and thus contribute to the diagnosis of angle-closure glaucoma.

*Fu et al. (2020b)* was the first to report the application of DL for this purpose. They developed a deep learning model using convolutional neural networks to detect angle closure in AS-OCT images from the Visante device. A comparison of deep learning algorithms with quantitative feature-based methods showed superior performance of the former, with an AUC value of 0.96 and proved a promising technology for the interpretation of OCT images. However, the usefulness of this system needs to be further validated in diverse population settings with the use of different devices (*Fu et al., 2020b*). In subsequent studies, the team refined the deep learning algorithm by incorporating several clinically relevant regions. They further evaluated its performance on AS-OCT images from the Cirrus device, achieving outstanding results (*Fu et al., 2020b*).

*Nongpiur et al. (2013a)* also aimed to classify primary angle-closure suspects (PACS) using AS-OCT and biometric parameters. This research involved a cross-sectional study of 243 PACS individuals. Hierarchical clustering and Gaussian Mixture Model (GMM) techniques were used to determine optimal subgroup classifications based on AS-OCT data. The analysis revealed three distinct PACS subgroups, characterized by differences in iris area, anterior chamber depth, and lens vault measurements. This subclassification enhances understanding of angle-closure pathogenesis and guides targeted therapeutic approaches (*Nongpiur et al., 2013a*).

AS-OCT imaging evaluates various angular parameters for assessing the anterior chamber angle. These parameters include angle open distance (AOD), anterior chamber width (ACW), and trabecular iris space area (TISA) (*Ni Ni et al., 2014*; *Nongpiur et al., 2017*). The screening, classification, and detection of anterior chamber angle closure through AS-OCT image analysis using deep learning methods are significant research areas. These approaches highlight the potential in glaucoma management (*Ting et al., 2021*).

*Xu et al. (2019)* developed a deep learning algorithm capable of detecting angle closure and PACD in all quadrants of AS-OCT images. The model demonstrated strong performance on the test dataset, achieving an AUC value of 0.928 for angle closure, and AUC values of 0.964 and 0.952 for PACD defined in two and three quadrants, respectively. These methods could be used to automate clinical evaluations of the ACA and improve access to eye care in high-risk populations (*Xu et al., 2019*).

To address challenges in scleral spur localization and angle closure classification, they subsequently released a large dataset of 4,800 labeled AS-OCT images from 199 patients. They also introduced an evaluation framework to standardize and compare various models. These advancements in artificial intelligence have the potential to drive further developments in AS-OCT image analysis, particularly in assessing angle-closure glaucoma (*Fu et al., 2020a*).

A recent study introduced and evaluated a 3D deep-learning-based digital gonioscopy system (DGS). This system is designed to identify narrow iridocorneal angles and peripheral anterior synechiae in eyes at risk of primary angle-closure glaucoma, using over 1.1 million images. The 3D DGS demonstrated high diagnostic accuracy, comparable to

that of ophthalmologists. This highlights its viability as a screening tool in primary eye care for individuals at high risk of PACG (*Li et al., 2022*).

*Randhawa et al. (2023)* assessed the generalizability and performance of a deep learning classifier for detecting gonioscopic angle closure in anterior segment OCT (AS-OCT) images. A convolutional neural network (CNN) model, developed with data from the Chinese American Eye Study (CHES), was tested across independent datasets from CHES, a community clinic in Singapore, and a hospital clinic at USC. The classifier showed consistent performance (AUC range: 0.890–0.932) and comparable agreement with human examiners (κ = 0.700–0.704). This automated method could support ophthalmologists in evaluating angle closure across diverse patient populations (*Randhawa et al., 2023*).

AI enhances AS-OCT imaging by utilizing extensive clinical and research data. It conserves time for both physicians and patients while improving diagnostic accuracy. AI technologies are advancing in their ability to identify significant eye diseases, such as diabetic retinopathy, primary open-angle glaucoma, visual impairment, and health risks in fundus images. Recently, there has been a shift from traditional machine learning (ML) techniques to the widespread use of deep learning (DL) algorithms. This shift has significantly increased the reports of using artificial intelligence to assist in the evaluation of primary angle closure disease (PACD) (*Balyen & Peto, 2019*; *Soh et al., 2024*).

The primary challenges in the traditional machine learning phase stem from the lack of convenient and accurate tools for clinical assessment and monitoring. The machine learning workflow includes several key steps: image pre-processing, identification of three PACD mechanism clusters (involving the iris, the lens, or a combination of components), feature extraction/selection, and classification. The goal of this workflow is to diagnose and grade glaucoma (*Baek et al., 2013*; *Kwon et al., 2017*; *Moghimi et al., 2018*).

Additional challenges to the development of artificial intelligence-based tools include scarcity of data and a lack of consensus in diagnostic criteria. Although research in the use of artificial intelligence for glaucoma is promising, additional work is needed to develop clinically usable tools (*Chen et al., 2023*).

In summary, current research is leading advancements in artificial intelligence and improvements in image acquisition technology. These developments aim to enhance the quality of research images and standardize imaging processes. As generative large language models and multimodal technologies continue to evolve, future glaucoma imaging research should focus on developing novel approaches. This includes employing multimodal techniques for the automatic measurement of multidimensional biometric parameters such as intraocular pressure, anterior chamber angle, visual field, fundus photos, and OCT. Additionally, there is a need to create intelligent analysis and early warning systems to better meet the demands of clinical practice, economic considerations, societal needs, and relevant policy frameworks (*Abràmoff et al., 2022*; *He et al., 2019*; *Teo et al., 2022*; *Topol, 2023*).

## Limitations

Bibliometrics is a crucial tool for evaluating scientific achievements; however, current analytical approaches may have limitations. First, we extracted only articles and reviews from 2004 to 2023, potentially overlooking other types of literature or significant publications from 2024. Secondly, the exclusive focus on English articles and reviews may lack comprehensiveness, excluding some pertinent publications. Lastly, downloading only the "full record and citations" from the WOS database may omit valuable details or insights. Although the WOS database is the most frequently utilized and recommended for bibliometric analyses, it may not cover all important publications. Therefore, future investigations should utilize other databases, such as PubMed or Scopus, to achieve a more comprehensive research scope.

## CONCLUSIONS

This study performed a bibliometric analysis of research on the application of AS-OCT in glaucoma diagnosis and treatment over the past 20 years. It identifies leading countries, institutions, authors, and research methods. The United States has made the most significant contributions in terms of article volume and citation impact. Among prolific authors, Aung Tin is the most notable with 77 articles and 3,428 citations. Since 2018, advancements in artificial intelligence have shifted the research focus from manual measurement to automated detection and recognition of relevant indicators. Future research will benefit from increased collaboration among ophthalmologists, physicists, and computer engineering experts.

## ACKNOWLEDGEMENTS

We would like to express our sincere gratitude to all the staff members of the Shenzhen Eye Hospital.

### Funding

This work was funded by the Shenzhen Fund for Guangdong Provincial High-level Clinical Key Specialties (No. SZGSP014), the National Nature Science Foundation of China (No. 82070961), the Shenzhen Key Medical Discipline Construction Fund (No. SZXK037), funded by the Shenzhen Science and Technology Program (No. JCYJ20220818103207015), the Sanming Project of Medicine in Shenzhen (No. SZSM202311012) and the Shenzhen Science and Technology Program (KCXFZ202307310933590040). The funders had no role in study design, data collection and analysis, decision to publish, or preparation of the manuscript.

### Grant Disclosures

The following grant information was disclosed by the authors:
Guangdong Provincial High-level Clinical Key Specialties: SZGSP014.
National Nature Science Foundation of China: 82070961.

Shenzhen Key Medical Discipline Construction: SZXK037.
Shenzhen Science and Technology Program: JCYJ20220818103207015,
KCXFZ202307310933590040.
Sanming Project of Medicine in Shenzhen: SZSM202311012.

## Competing Interests

The authors declare that they have no competing interests.

## Author Contributions

- Yijia Huang conceived and designed the experiments, performed the experiments, analyzed the data, prepared figures and/or tables, authored or reviewed drafts of the article, and approved the final draft.
- Di Gong conceived and designed the experiments, analyzed the data, authored or reviewed drafts of the article, and approved the final draft.
- Kuanrong Dang conceived and designed the experiments, analyzed the data, authored or reviewed drafts of the article, and approved the final draft.
- Lei Zhu performed the experiments, analyzed the data, prepared figures and/or tables, authored or reviewed drafts of the article, and approved the final draft.
- Junhong Guo performed the experiments, analyzed the data, prepared figures and/or tables, authored or reviewed drafts of the article, and approved the final draft.
- Weihua Yang conceived and designed the experiments, authored or reviewed drafts of the article, and approved the final draft.
- Jiantao Wang conceived and designed the experiments, authored or reviewed drafts of the article, and approved the final draft.

## Data Availability

This is a literature review.

## Supplemental Information

Supplemental information for this article can be found online at http://dx.doi.org/10.7717/peerj.18611#supplemental-information.

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
