# Peer review of "The applications of anterior segment optical coherence tomography in glaucoma: a 20-year bibliometric analysis"

_PeerJ, doi:10.7717/peerj.18611_

## Round 0.1 · original submission · Major Revisions

· Academic Editor

Major Revisions

The reviews range from Accept (R3) to major Revisions (R2). Please address all concerns, and this submission will then be re-evaluated

Reviewer 1 ·

Basic reporting

1- Introduction is not sufficient regarding the domain of review and what will added
2- Please clarify inclusion criteria among studied articles
3-Please added more details in artificial intelligence section with more clinical applicatinos

Experimental design

This is a retrospective systematic review with meta-analysis study

Validity of the findings

well done but conclusion should be condensed

Additional comments

Please clarify abbreviations, authors contributions, standards of reporting, ORCID of authors

Reviewer 2 ·

Basic reporting

1、Language and Grammar:
Overall Clarity: It is recommended to proofread the entire text to ensure clarity and professionalism in language. For example, sentences like "By 2020, the number of patients with POAG and PACG is estimated to reach" need to be reorganized for improved readability and accuracy. Consider having an English professional familiar with medical terminology proofread the document.
Specific Phrases: The wording in sentences on lines 23, 77, 121, and 128 currently makes it difficult to understand. Please revise these sentences to enhance clarity and conciseness.

2、Introduction and Background:
This article provides valuable insights into the application of AS-OCT in glaucoma research. However, significant revisions are needed in terms of clarity, methodological rigor, and comprehensiveness of literature citations to meet high academic standards.

Experimental design

1、Statistical rigor: Ensure that statistical analysis is rigorous and appropriate for the data. Provide rationale for selecting these analytical methods, particularly utilizing the methods of VOS viewer and CiteSpace.

2、Reproducibility: Methods should be described in detail so that other researchers can replicate the study. This includes search strategies, data extraction processes, and analysis techniques.

Validity of the findings

1、Relevance to the original research question: Ensure that the conclusions are directly related to the research question proposed in the introduction. Provide a detailed discussion on the implications of the research findings, considering their impact on future research and clinical practice.

2、Unresolved issues: Identify any unresolved issues or gaps that need to be addressed in future research. This will help drive the development of the field and guide new research directions.

Additional comments

NA

·

Basic reporting

No comment

Experimental design

No comment

Validity of the findings

No Comment

Additional comments

I think there might be some interest in the ethnic base of the many studies. Is it possible to analyse the data to fully separate European and Asian populations? It would be useful to know which of the US articles are strongly biased in favour of recruitment of white European ethnic groups. Also do any of the studies include a substantial group of African/African origin patients?

Otherwise I found this a well written and fascinating study, of great value to any researcher in the area. However I am not familiar with the methodology and so am not able to comment on the analyses carried out

---

## Round 0.2 · accepted · Accept

· Academic Editor

Accept

After revisions, all reviewers agreed to publish the manuscript. I also reviewed the manuscript and found no obvious risks to publication. Therefore, I also approved the publication of this manuscript.

Reviewer 1 ·

Basic reporting

well done

Experimental design

good

Validity of the findings

good

Additional comments

The article in revised form is ready forv publications.

Reviewer 2 ·

Basic reporting

Revised manuscript was improved and satisfied.

Experimental design

Revised manuscript was improved and satisfied.

Validity of the findings

Revised manuscript was improved and satisfied.

Additional comments

NA